**Data Availability Statement:** All relevant data are within the manuscript and its Supporting Information files.

# The SIESTA (SEAAV Integrated evaluation sedation tool for anaesthesia) project: Initial development of a multifactorial sedation assessment tool for dogs

Fernando Martinez-Taboada[1]☯*, Jose Ignacio Redondo[2]☯

**1** Department of Veterinary Anaesthesia and Analgesia, School of Veterinary Science, Faculty of Science, The University of Sydney, Camperdown, New South Wales, Australia, **2** Departamento de Medicina y Cirugía Animal, Facultad de Veterinaria, Instituto de Ciencias Biomédicas, Universidad CEU Cardenal Herrera, Valencia, Spain

☯ These authors contributed equally to this work.
\* fer_m_taboada@hotmail.com

## Abstract

### Objective

The aim of the study was to develop a multifactorial tool for assessment of sedation in dogs.

### Methods

Following a modified Delphi method, thirty-eight veterinary anaesthetists were contacted to describe the following levels of awareness: no-sedation, light, moderate, profound sedation and excitation. The answers were summarized in descriptors for each level. A questionnaire was created with all the variables obtained from the descriptors. The questionnaire was returned to the panel of anaesthetists to be used before and after real sedations in conjunction with the previous 5-point categorical scale. Data obtained were analysed using the classification-tree and random-forest methods.

### Results

Twenty-three anaesthetists (60%) replied with descriptions. The descriptors and study variables were grouped in categories: state-of-mind, posture, movements, stimuli-response, behaviour, response-to-restraint, muscle tone, physiological data, facial-expression, eye position, eyelids, pupils, vocalization and feasibility-to-perform-intended-procedure. The anaesthetists returned 205 completed questionnaires. The levels of awareness reported by the anaesthetists were: no sedation in 92, mild (26), moderate (37) and profound in 50 cases. The classification-tree detected 6 main classifying variables: change in posture, response-to-restraint, head-elevation, response-to-toe-pinching, response-to-name, and movements.

The random-forest found that the following variables: change in posture, response-to-restraint, head-elevation, response-to-name, movements, posture, response-to-toe-

**Funding:** The author(s) received no specific funding for this work.

**Competing interests:** The authors have declared that no competing interests exist.

pinching, demeanour, righting-reflex and response-to-handclap, were classified correctly in 100% awake, 62% mild, 70% moderate and 86% of profound sedation cases.

## Discussion and conclusion

The questionnaire and methods developed here classified correctly the level of sedation in most cases. Further studies are needed to evaluate the validity of this tool in the clinical and research setting.

## Introduction

Sedation is a state characterized by central depression accompanied by drowsiness and some degree of centrally induced relaxation [1]. The term is very broad and it is regularly used to refer to a range from a calm and stress-free state required just to tolerate hospitalization, to a much more profound depression of the central nervous system with immobility and no response to painful stimulus. Even, in recent years, the use of procedural sedation as an alternative to general anaesthesia is becoming popular for minor surgical and diagnostic procedures.

The assessment of the degree of sedation experienced by animals is fundamental both in clinical practice and in research. This assessment can enhance the accurate titration of anaesthetic agents to reduce the incidence of excessive drug-induced complications [2]. It can also be a fundamental tool in the development of new sedative drugs, drug combinations and routes of administration.

Clinicians and researchers require tools to measure the effectiveness of sedation in individual animals in relation to the purpose of such sedation. These tools are the sedation scales or scoring systems [3]. Different types of sedation scales have been previously published: numerical [4], simple descriptive [5] and multifactorial [6]. To the best of the authors' knowledge, there is no consensus in the veterinary anaesthesia community regarding the best sedation tools to be used clinically or in research. Additionally, only one study has been published attempting to preliminarily validate a canine sedation scale [7].

The aim of this work was to collect and classify veterinary anaesthetists' opinions on canine sedation characteristics and behaviours, and to develop a tool to assess the level of sedation.

## Material and methods

A modified Delphi method was used to obtain the opinions of anaesthetists working in clinical practice [8]. The study began in February 2016 and ended in April 2016 and it was approved by the Universidad CEU Valencia Human Research Ethics Authority (no 2012–813). For the last phase, the Animal Research Ethics Committee advised that specific ethics approval for this observational study was not required as the study design did not cause deviation from standard practice.

### Experts/Participants

Thirty-eight full-time veterinary anaesthetists were contacted to explore their interest to be part of the expert panel. They all spoke Spanish as their mother language, were members of the SEAAV (Sociedad Española de Anestesia y Analgesia Veterinaria, Spanish for Spanish Society of Veterinary Anaesthesia and Analgesia), and had postgraduate training in veterinary

anaesthesia and analgesia (diploma of the European College of Veterinary Anaesthesia and Analgesia [ECVAA], diploma of the American Colleges of Veterinary Anaesthesia and Analgesia [ACVAA], completion of an ECVAA or ACVAA approved residency program, PhD, MSc, etc.).

The experts were contacted by email to explain the aims of the study and to give some general information regarding the study. Those who replied agreeing to participate and providing consent ('participants' for the rest of the text) were included in a mailing list. This email mailing list was used in all the subsequent communication phases of the study.

## Consultation process

The consultation process was subdivided in 3 phases or rounds.

Phase 1: Following the Delphi method, a group of broad open questions were formulated. The participants had to describe, in their own words, the following categories: no sedation, light sedation, moderate sedation, profound sedation, and excitation.

The answers were visually analysed in the form of word clouds to graphically assess the most frequent or relevant expressions. The answers were summarised and structured in descriptors that were shared with the group of participants for their feedback.

Phase 2: A list of dichotomic variables were developed from the descriptors obtained in phase 1 and a preliminary questionnaire was developed. The list of descriptors and the preliminary questionnaire were circulated among the participants to obtain their opinion regarding total or partial repetition of variables, clarity of the variables to be assessed and overall usability of the questionnaire.

Phase 3: With the information obtained from the previous phases, a definitive questionnaire was developed as a fillable portable document format (pdf) (Adobe Acrobat X Pro, Adobe Systems, San Jose, CA, USA). This questionnaire (Appendix 1) included 10 sections: Date and assessor details, animal signalment and procedure, sedation protocol (including drugs, doses and route of administration), a group of variables related to the dog's state of mind and posture (alertness, mental and emotional state, head position, posture, righting reflex and general behaviour prior to interaction with the dog), variables related to mobility (eg. muscle tone, ataxia, hypermetric movements amongst others), a group of miscellaneous variables obtainable by passive observation of the animal (eye position, pupil size, visible third eyelid, presence of nystagmus, excessive salivation, vocalization, respiratory rate and panting), variables related to the dog's response to different types of stimuli (eg. menace response, response to a handclap, response to being called by their name, response to holding one of the front paws, etc), variables in relation to the feasibility of performing the intended procedure or step (hair clipping, venous catheter placement, arterial catheter placement, radiographical studies, preparation of the surgical field), the five-point simple descriptive sedation scale (no-sedation, light, moderate, profound sedation and excitation) used in phase 1, and a free text box for any other observations. During the phase 2 feedback, it was made clear that some variables could not be assessed in every sedation status. For this reason, the variables related to behaviour and some of the ones related to movements, were optional variables that appeared with a tick box next to them. All the other variables required a compulsory answer and they appeared with a circle next to them. During this 3rd phase, the participants were asked to use the questionnaire on clinical cases within their practice. They had to assess at least five cases before the administration of any sedative drugs and another five cases after the administration of sedative drugs (the assessments could be done on the same dogs before and after sedation, but they could also be done on different animals). The entire questionnaire needed to be completed, including the subjective five-point simple descriptive sedation scale. Partially completed forms were not allowed to be submitted.

The participants had a week to reply for phase 1, the same period for phase 2, and a two-week period for phase 3. The communication between participants and coordinators was always in Spanish. All the data provided by the participants was compiled, summarized and deidentified by the coordinators before being returned for their assessment and feedback.

The responses to the descriptors in the questionnaires were analysed in conjunction with the score provided by the participants using the five-point simple descriptive scale.

## Data analysis

The data obtained from the questionnaires was analysed with the statistical program R 3.5.3 (R Core Team 2019, The R Foundation for Statistical Computing http://www.R-project.org, Austria) performing a classification tree using the function rpart() of the statistical package rpart [9]. The minimum number of observations per node to try a partitioning was set at 15. The classification tree was plotted via the function rpart.plot() of the rpart.plot package [10].

The data was also analysed using the randomForest package [11,12]. This package randomly selected 66% of the results as a learning sample and built a classification tree with this data. In this case, it was considered that each node should have a minimum of 5 observations. The rest of the data (33% of the results) was considered out-of-bag data and it was used to assess the sensitivity and specificity of the classification tree. This process of generating a classification tree and to check its classification functions was repeated 5000 times. The weight of each category or sedation status (no-sedation, light, moderate, profound sedations and excitation) was pondered according to relative percentage of frequency of analysed cases, so the initial probability of any given dog to be scored in any of the categories was 20%.

Data related to the animal signalment, procedure and sedation protocols was analysed by descriptive statistics and is presented in the form of mean ± SD, number of observations and/or percentage.

## Results

A total of 23 of the 38 contacted participants (60%) consented to participate in the study.

The descriptors obtained to the open questions asked in phase 1 were classified into the following 15 groups: state of mind, spontaneous posture, mobility, response to stimuli, response to visual presence, behaviour, response to restraint, muscle tone, physiological data, facial expression, eye position, eyelids position, pupils (presence of mydriasis), vocalization and feasibility to perform the intended procedure.

The final version of the questionnaire (annex 1) included 106 dichotomic variables.

The participants returned 205 completed forms. A total of 94 male and 101 female dogs, 5.2 ± 3.8 year-old and weighing 18.4 ± 13.0 kg were assessed with the questionnaire. Only 10 dogs (4 male and 6 female) were assessed twice, before and after the sedation. The rest of the dogs were only assessed once, either fully awake (before any sedation was administered) or after being sedated. Their health status was classified as ASA I in 92 dogs (44.9%), ASA II in 83 cases (40.5%) and ASA III in 30 (14.6%). No animals were classified ASA IV or V. The reasons for sedation were: pre-anaesthetic sedation (68%), diagnostic procedures (28%) and others (4%).

The level of sedation, assessed by the participants using the 5-point scale, was no sedation in 92 cases (44.9%), mild sedation in 26 cases (12.7%), moderate sedation in 37 cases (18%) and profound sedation in 50 (24.4%). There were no cases classified as "excitation". The drugs used are shown in Table 1.

The classification tree obtained (Fig 1) shows in each node the classifying descriptor or variable that allows the dichotomization of the cases. According to this classification tree, six

**Table 1.**

| Sedatives | | | Analgesic | | |
|---|---|---|---|---|---|
| **Drug name** | **No cases** | **%** | **Drug name** | **No cases** | **%** |
| Acepromazine | 24 | 11.7 | Methadone | 72 | 35.1 |
| Medetomidine | 24 | 11.7 | Morphine | 12 | 5,9 |
| Dexmedetomidine | 66 | 32.2 | Pethidine | 7 | 3.4 |
| Midazolam | 7 | 3.4 | Fentanyl | 7 | 3.4 |
| Diazepam | 1 | 0.5 | Butorphanol | 14 | 6.8 |
| | | | Buprenorphine | 2 | 1 |
| | | | NSAIDs | 9 | 4.4 |

Drugs used for sedation by the participants with the number of cases used for and the percentage. Please note that in multiple cases, the reported combination contained more than one sedative and one analgesic.

predictors overtook all the other explanatory variables: change in posture, head-elevation, response-to-restraint, response-to-name, movements and posture.

The random forest determined the most important variables for a successful classification were (Fig 2): change in posture, response to restraint, head elevation, response to name, movements, posture, response to toe pinching, demeanour, righting reflex and response to handclap based on the variable important plot. And based on the mean decrease in Gini coefficient, the most important variables were: head-elevation, response-to-name, change in posture, response-to-restraint, posture and movements (Fig 2).

The final random forest model predicted accurately the different levels of sedation (Table 2). The out of bag estimate of error rate was 14.15%.

## Discussion

The Delphi technique was a successful method to obtain a collective view in an area of very limited published data. This technique has proven adequate in bringing consensus in areas of knowledge with no or limited evidence and that are very dependent on individual opinion [13].

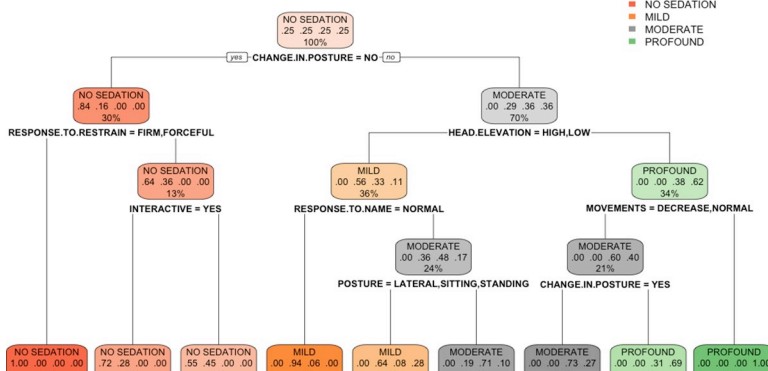

**Fig 1. Classification tree algorithm.** The classification tree represents the different selection criteria or 'decision nodes' used to predict the most correct classification of the total number of cases (represented at the root of the tree as a 100%). As the data is classified in subsets, the percentage value represents the probability of a case of belonging to that data subset.

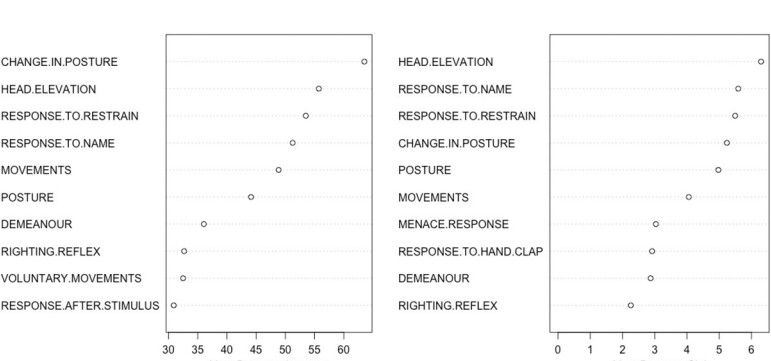

**Fig 2. Variable importance plot (mean decrease accuracy and mean decrease Gini).** This is a fundamental outcome of the random forest and it shows, for each variable, how important it is in classifying the data. The Mean Decrease Accuracy plot expresses how much accuracy the model losses by excluding each variable. The more the accuracy suffers, the more important the variable is for the successful classification. The variables are presented from descending importance. The mean decrease in Gini coefficient is a measure of how each variable contributes to the homogeneity of the nodes and leaves in the resulting random forest. The higher the value of mean decrease accuracy or mean decrease Gini score, the higher the importance of the variable in the model.

The Delphi technique requires a panel of 'experts' and in this research 23 participants formed that panel. The definition of 'expert' was deliberately ambiguous. The only pre-requisites for the being 'experts' were that the participants were working full time in veterinary anaesthesia and they held some postgraduate qualification in veterinary anaesthesia. These requirements were set at that level for two reasons: 1. Easy communication between the panel of participants and the team of coordinators (eg. using the same terminology and jargon) and 2. The participants being able to readily use the questionnaire when developed. Theoretically, the panel size can vary from 4 to 3000, and the final number is just an empirical and pragmatic decision [14]. The coordinators and authors of this research discussed the panel size and a bigger number of participants with smaller work-load per individual was considered preferable to maintain a high level of engagement from the participants. The use of electronic communication and very strict deadlines for each step of the process allowed the completion of the Delphi technique in this study in just under six weeks, compared to several months in similar studies [15].

The classification and regression trees are techniques to determine relations between multiple independent variables and a dependent one. The term classification tree is used when it is applied to discrete or categorical variables (like in this case) and regression trees when the

**Table 2.**

| | | Predicted scores | | | | | |
|---|---|---|---|---|---|---|---|
| | | No sedation | Mild | Moderate | Profound | Agreement | Class error |
| **Observed scores** | **No sedation** | 92 | 0 | 0 | 0 | 100.0% | 0.0% |
| | **Mild** | 6 | 16 | 4 | 0 | 61.5% | 38.5% |
| | **Moderate** | 0 | 5 | 26 | 6 | 70.3% | 29.7% |
| | **Profound** | 0 | 0 | 7 | 43 | 86.0% | 14.0% |

Confusion or error matrix showing the observed and predicted (based on the random forest model) values, agreement (proportion of data subsets predicted correctly by the model) and the class or classification error (proportion of data subsets predicted wrongly by the model).

variables are continuous. Both techniques were described by Breiman et al. [16] as an algo-
rithm that is dividing the data into subgroups to minimise the heterogeneity (called node
impurity) within the different subgroups. This technique always classifies the variables by
importance [17].

The classification tree determined that the most relevant variables for the classification of
sedation were: change in posture, head elevation, response to restraint, response to name,
movements and spontaneous posture. The random forest determined the most useful variables
were: change in posture, response to restraint, head elevation, response to name, movements,
posture, response to toe pinching, demeanour, righting reflex and response to handclap. Some
of these domains coincide with those previously mentioned in other composite sedation scales
used in the literature. Young et al. assessed the level of sedation after medetomidine by analys-
ing six parameters: jaw tone, placement on side, response to noise, attitude, posture and pedal
reflex [18]. These variables are mostly included in the ones obtained in this study with the
exception of jaw tone. The assessment of jaw and tongue tone also appear in another two com-
posite sedation scales in dogs [19, 20]. This variable was not mentioned in any of the phases of
the Delphi method reported here. It was not even a descriptor mentioned in phase 1 as an
answer to the open questions, it was not mentioned during the phase 2 feedback and, for that
reason, it was not included in the questionnaire. It is possible that the participants favoured
other methods for the assessment of this physical characteristic such as the animal's muscle
tone. It is also possible that the participants strongly associated the evaluation of jaw tone with
the level of anaesthetic depth and this might have prevented them from relating it to sedation.
This explanation would seem unlikely considering that they included the position of the eye
and the palpebral reflex as adequate variables to assess sedation, which are also used widely to
assess depth of anaesthesia. The previously published scales also consider the position of the
eye and the presence of palpebral reflex [19,20], but, although these variables were included in
our questionnaire, neither the classification tree or the random forest found them determinant
in the assessment of sedation. Wagner et al. [7] also observed that they could shorten their
scale by removing three domains (palpebral reflex, jaw tone and resistance when laid into lat-
eral recumbency) maintaining similar consistency to the full scale, but being three times
shorter to complete. In the research reported here, the righting reflex was not significant in the
classification tree, but it was a significant domain in the random forest analysis. Additionally,
the head elevation was found to be fundamental in the classification of the different levels of
sedation. Head elevation is a variable that it has never been mentioned in the previously pub-
lished sedation scales in small animals (although it is a variable extensively used in horses
[21]).

The random forest analysis classified correctly all the non-sedated animals and 43 out of 50
(86%) animals profoundly sedated, when compared with the participants' assessment using
simple descriptive scales. The success rate in the classification decreases when analysing mild
(61.5%) and moderate (70.3%) sedations. The degree of central depression obtained after the
administration of sedatives is a continuous variable with several physical signs. As a conse-
quence, the assessment of this depression is very subjective. In this research, the simple
descriptive scale was deliberately open to the participants' interpretation. No descriptors for
the different categories were provided, so it is possible that there was some degree of overlap-
ping between different participants and categories. In other words, it is possible that a mildly
sedated dog for an observer might have been moderately sedated for a different participant.
This might explain some of the classification errors committed by the random forest analysis.

Composite rating scales are multi-item scales that allow detailed evaluation of complex or
multidimensional issues. Several items referring to the same idea or issue have to increase
internal consistency and, consequently, they should improve the reproducibility and validity

of the scale [22]. The variables used in the present study were always assessed and scored in a dichotomous (yes/no) answer. This type of answer counteracts any subjectivity in the assessment of any variable. For example, Grint et al. [20] developed and used a composite scale for assessing sedation levels in dogs that was later validated by Wagner et al. [7]. This scale allows each domain to be assessed in 3 or 4 possible gradings (associating a score from 0 to 2 or 3 depending of the grading). These types of simple descriptive scales within the different domains allows extensive assessor subjectivity in the interpretation of each subsection of the composite scale.

It is important to comment on several limitations regarding this study. All the participants knew when the dogs were non-sedated, as they performed the assessment before administering drugs to the animals. However, it is impossible to predict the effect of any drug combination in a particular individual, so the responses for the sedated dogs should be accurate. Inter-observer and intra-observer variability over time were not studied, as the assessments were performed *in situ* by the participants. The time to complete the questionnaire was not assessed, but presumably this was long. The original intention was to identify important variables for the classification of sedation and then, to develop a shorter questionnaire. For that reason, the original inventory was long and comprehensive. Finally, the Delphi method requires a group of implicated and flexible facilitators/coordinators that can classify, compile and condense the group's ideas to streamline them and to avoid repetition. There is no doubt the coordinator group could have introduced some degree of bias during this process, but the Delphi method performed in this study involved three rounds and, after each of them, the responses were aggregated and shared for feedback from the participants. It is fairly unlikely, that the coordinators could have forced any bias in the research without being noticed by the participants.

## Conclusion

The Delphi method and the random forest analysis identified ten domains to successfully classify most of the levels of sedation in the dog. Further studies are now needed to assess the validity of the short version of the questionnaire and its reliability. Additionally, a broad study using a large number of dogs and veterinarians with and without specific anaesthesia knowledge would also allow the assessment of the tool agreement and bias.

## Supporting information

**S1 Appendix. Questionnaire.** Final version of the questionnaire distributed to the collaborators.
(PDF)

**S1 Data.**
(PDF)

**S2 Data.**
(PDF)

**S3 Data.**
(PDF)

## Acknowledgments

Special thanks to all the participants for their generous and selfless support.

## Author Contributions

**Conceptualization:** Fernando Martinez-Taboada, Jose Ignacio Redondo.

**Data curation:** Fernando Martinez-Taboada, Jose Ignacio Redondo.

**Formal analysis:** Jose Ignacio Redondo.

**Investigation:** Fernando Martinez-Taboada, Jose Ignacio Redondo.

**Methodology:** Fernando Martinez-Taboada, Jose Ignacio Redondo.

**Project administration:** Fernando Martinez-Taboada.

**Supervision:** Fernando Martinez-Taboada, Jose Ignacio Redondo.

**Writing – original draft:** Fernando Martinez-Taboada.

**Writing – review & editing:** Fernando Martinez-Taboada, Jose Ignacio Redondo.

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
