## [Decision Letter · Decision Letter 0]

27 Sep 2019

PONE-D-19-17102

The SiESTa (SEaav SedaTion) Scale project: development of a multifactorial composite sedation inventory for dogs

PLOS ONE

Dear Dr Fernando Martinez-Taboada,

Thank you for submitting your manuscript to PLOS ONE. After careful consideration, we feel that it has merit but does not fully meet PLOS ONE’s publication criteria as it currently stands. Therefore, we invite you to submit a revised version of the manuscript that addresses the points raised during the review process.

Dear Authors, both reviewers expressed some concerns regarding your manuscript. In particular the second reviewer had the major concerns and some of them are very critical. I invite you to try to respond accordingly to all the concerns of the reviewer and amend the manuscript accordingly

We would appreciate receiving your revised manuscript by Nov 11 2019 11:59PM. To enhance the reproducibility of your results, we recommend that if applicable you deposit your laboratory protocols in protocols.io, where a protocol can be assigned its own identifier (DOI) such that it can be cited independently in the future. For instructions see: http://journals.plos.org/plosone/s/submission-guidelines#loc-laboratory-protocols

We look forward to receiving your revised manuscript.

Kind regards,

Francesco Staffieri

Academic Editor

PLOS ONE

Journal Requirements:

2.  Please include additional information regarding the survey or questionnaire used in the study and ensure that you have provided sufficient details that others could replicate the analyses. For instance, if you developed a questionnaire as part of this study and it is not under a copyright more restrictive than CC-BY, please include a copy, in both the original language and English, as Supporting Information."

3. Please amend your current ethics statement to include the full name of the ethics committee that approved your specific study.

For additional information about PLOS ONE submissions requirements for animal ethics, please refer to http://journals.plos.org/plosone/s/submission-guidelines#loc-animal-research  

Once you have amended this/these statement(s) in the Methods section of the manuscript, please add the same text to the “Ethics Statement” field of the submission form (via “Edit Submission”)

Additional Editor Comments (if provided):

Dear Authors, both reviewers expressed some concerns regarding your manuscript. In particular the second reviewer had the major concerns and some of them are very critical. I invite you to try to respond accordingly to all the concerns of the reviewer and amend the manuscript accordingly

Reviewers' comments:

Reviewer's Responses to Questions

**Comments to the Author**

1. Is the manuscript technically sound, and do the data support the conclusions?

Reviewer #1: Partly

Reviewer #2: No

2. Has the statistical analysis been performed appropriately and rigorously? 

Reviewer #1: Yes

Reviewer #2: I Don't Know

3. Have the authors made all data underlying the findings in their manuscript fully available?

Reviewer #1: Yes

Reviewer #2: No

4. Is the manuscript presented in an intelligible fashion and written in standard English?

Reviewer #1: Yes

Reviewer #2: No

5. Review Comments to the Author

Reviewer #1: the acronym used is not consistent with the words: I suggest to find a single word for each letter for example SIESTA dog:Seaav Integrated Evaluation Sedation Tool for Anaesthesia in dogs

Abstract

Line 45: please specify how many anaesthetists sent the questionnaires back. It is not clear how the clinicians involved used the questionnaires.

Line 56: there are not enough information in the abstract to asses that the questionnaire used was successful in evaluating the real sedation status of the animals.

M&M

Line 113: please specify if the answer to this first phase was anonymous or not.

Results

Line 184: please correct the capital letter of “State” in “state”.

Line 191: please specify why on 205 case there were 94 male and 101 female: 10 dogs are missing.

Table 1: I suggest to put the total number for each class of drugs in order to see what is explained in the legend: more than 1 drug was used on the same animal. Anyway is not clear why the authors want to give this information because is not possible from the table understand which kind of sedative protocol/combination was done; I suggest to divide the cases in only opioid, or only sedative agents and combinationS (opioids+ sedatives or sedative + tranquillisers, etc.). It would be interesting to associate the drugs used to the level of sedation, to see if the evaluation of sedation was consistent with the protocol.

Unfortunately the annexes are in spanish and not in english.

In M&M is missing the description of comparison between the level so sedation given by the descriptive scale and the one resulted from the questionnaire.

Reviewer #2: Reviewer report

This manuscript describes the attempted development of a sedation scale in dogs, based on a Delphi method.

I have limited my comments to the study design and underlying assumptions as the analytical techniques (classification tree and random forest) employed are beyond my expertise.

I have concerns regarding the study methods, some of which may profoundly limit interpretation of the results.

1. Study participants - participants are repeatedly described as “experts” but there is no evidence, based on information provided in the manuscript, how many of them meet widely accepted definitions of expert status (American or European College Diplomate holders). Holding a research degree (MSc or PhD) is not a reflection of clinical experience or training, or even relevance to the subject of this study. This poses several problems, some of which are likely to have interfered with the collected data and consequently biased the results and interpretation.

a. participants may have provided limited descriptors, reflecting limited training experience. To some extent this is reflected in the Discussion, where the authors identify that some criteria, such as jaw tone, were not mentioned.

b. participants reflect a single mother tongue, Spanish. It is well established that health assessment scale development should consider differences in the interpretation and meaning of terms between languages. It is concerning that a scale developed in Spanish has been translated in to English for publication, with no apparent assessment of whether scale performance has been affected as a result of translation.

c. Participant assessments using the developed scale were compared to a simple descriptive scale. This problematic because feedback from the same participants was used to develop the scale. Ideally, animals would have been video-recorded (with appropriate consent) and the the videos scored by a separate, independent group of observers.

d. the authors suggest that similarities in terminology and jargon used by participants during the Delphi process were beneficial. Given the lack of apparent minimal consistent training level, how could authors be certain that terminology and jargon were applied consistently by participants?

2. Scale development process

a. the argument in favour of dichotomous outcomes as being more objective is misguided. Though forcing a participant to select a dichotomous outcome appears objective, the underlying assessments are in large part subjective. There is no indication that objective measures were employed in many of the final predictors described. This could explain why the scale performed less well when animals had levels of sedation that were not at the extremes (“no” or “profound” sedation).

b. from the methods (phase 1 and 2) it appears there was considerable author input in selecting and classifying the final descriptors - this may be a necessary step, but it would be helpful to have assurance that any author bias was limited - perhaps the authors would consider making their data available on-line?

c. some descriptors are clearly highly subjective and uncontrolled e.g. responses to noise and name.

d. for some descriptors, participants had the option of selecting which would be tested. This suggests the possibility that the testing method was not standardised, with the risk that the outcome of a test would be influenced by what occurred previously. For example, if attempts were made to place a dog in lateral recumbency, this could change the level of sedation, affecting the test that followed (e.g. response to noise).

e. There appears to be an underlying assumption that the method used by the authors to collect descriptors is somehow superior to methods used in previously published studies (e.g. Young et al.). Just because the described method has been named “Delphi” does not confer legitimacy over older studies conducted before the Delphi concept existed or was widely applied. It is highly likely that descriptors generated in previous studies were achieved through consensus discussion.

f. I could not find Fig 2.

g. As identified by the authors, participants were not blinded to the status of the dogs. This would appear a critical limitation, especially when no further testing has been done to evaluate scale performance with observers blinded to treatment.

3. The language used throughout needs editing to raise the standard of scientific english. There is a tendency to use slang (“in fact”) and poorly defined terms (“awareness” - how can we know if an animal is aware? Similarly, “state of mind”).

4. As the authors describe, other sedation scales have been published for use in dogs, including one that was validated according to psychometric principles (validity and reliability testing). With the study goal of developing this novel scale, it would have been invaluable to perform some comparisons with pre-existing scale. In general terms, it is extremely difficult to interpret the performance of a novel scale when it is compared against a non-validated scale (as was the case here). Therefore, it is probably unsurprising that the results show the extremes of sedation (no sedation and profound sedation) to be in close agreement. Surely it is accurately classifying the full continuum (including crossing the threshold in to general anaesthesia) that is of greatest interest?

6. PLOS authors have the option to publish the peer review history of their article (what does this mean?). If published, this will include your full peer review and any attached files.

Reviewer #1: No

Reviewer #2: No

---

## [Author Response · Author response to Decision Letter 0]

11 Nov 2019

To the reviewer 1:

Thank you for taking the time to review our work. We truly believe in the peer-review process and we know the article will improve with your comments. All the modifications that you suggested have been highlighted yellow in the manuscript. The reply to your comments is as follow:

You mentioned in your revision:

the acronym used is not consistent with the words: I suggest to find a single word for each letter for example SIESTA dog:Seaav Integrated Evaluation Sedation Tool for Anaesthesia in dogs

Although there is no need for an acronym to use the first letter of the words, we appreciate your suggestion and we have decided to introduce it in the manuscript. 

Abstract

Line 45: please specify how many anaesthetists sent the questionnaires back. It is not clear how the clinicians involved used the questionnaires.

An explanation has been added in line 45 and the number of participants are included in line 48.

Line 56: there are not enough information in the abstract to asses that the questionnaire used was successful in evaluating the real sedation status of the animals.

We added some modification to the abstract to address this comment.

M&M

Line 113: please specify if the answer to this first phase was anonymous or not.

This has been added now.

Results

Line 184: please correct the capital letter of “State” in “state”.

This has been changed now.

Line 191: please specify why on 205 case there were 94 male and 101 female: 10 dogs are missing.

Information to clarify this has been added now. These animals were assessed twice and for that reason they appeared missing.

Table 1: I suggest to put the total number for each class of drugs in order to see what is explained in the legend: more than 1 drug was used on the same animal. Anyway is not clear why the authors want to give this information because is not possible from the table understand which kind of sedative protocol/combination was done; I suggest to divide the cases in only opioid, or only sedative agents and combinationS (opioids+ sedatives or sedative + tranquillisers, etc.). It would be interesting to associate the drugs used to the level of sedation, to see if the evaluation of sedation was consistent with the protocol.

We are providing the information only because it was included in the questionnaire and we thought that it will give some degree of perspective to the reader and for completeness. We agree, the information is fairly irrelevant and we defer to the Editor to decide if this should be included or not.

Unfortunately the annexes are in spanish and not in english.

As mentioned in the Materials and Methods, all the research was performed in Spanish and for that reason the data in the annexes is all in Spanish. Any translation will involve some degree of interpretation and we are considering the possibility of validating this tool in other languages, but as yet this research is just in the initial development phase. For that reason, we have kept the translation to a minimum for manuscript publication.

In M&M is missing the description of comparison between the level so sedation given by the descriptive scale and the one resulted from the questionnaire.

This has now been added in line 149.

We hope these comments answer all your queries and the changes in the manuscript address your concerns.

Once again, thank you for reviewing our work.

The authors.

 

To the reviewer 2:

Thank you for taking the time to review our work. We truly believe in the peer-review process and we know the article will improve with your comments. All the modifications that you suggested have been highlighted green in the manuscript. The reply to your comments is as follow:

You mentioned in your revision:

This manuscript describes the attempted development of a sedation scale in dogs, based on a Delphi method.

We think it is important to highlight that this study is the development of a sedation assessment tool. The title has been changed to stress that. We were trying to define and apply a new approach to the question about how sedated the dogs are. We decided to approach the question from a different angle, with a group of questions that when they are answered in a simple way (yes/no) and in a specific order, the level of sedation can be easily defined. We believe the term “scale” brings some preconceived ideas and, for that reason, we have removed it from the manuscript as much as possible.

I have limited my comments to the study design and underlying assumptions as the analytical techniques (classification tree and random forest) employed are beyond my expertise.

I have concerns regarding the study methods, some of which may profoundly limit interpretation of the results.

1. Study participants - participants are repeatedly described as “experts” but there is no evidence, based on information provided in the manuscript, how many of them meet widely accepted definitions of expert status (American or European College Diplomate holders). Holding a research degree (MSc or PhD) is not a reflection of clinical experience or training, or even relevance to the subject of this study. This poses several problems, some of which are likely to have interfered with the collected data and consequently biased the results and interpretation.

As it is mentioned in the manuscript (Line 273 – ‘The definition of ‘expert’ was deliberately open to people with various levels of experience and expertise, independently of the qualifications. The only requirement was that the participants were working full time in veterinary anaesthesia’. By definition, an expert is a knowledgeable or skillful person in a specific topic. We are sure that you agree with us that the people capable to assess sedation in dogs are the ones who are performing that task every day. We deliberately stayed away from the conservatism of considering only ECVAA/ACVAA diploma holders as ‘experts’ because some of them/us do not practice clinically as much as other people considered ‘less expert’ by that conservative definition. For your peace of mind, we can tell you that of the 23 participants 10 hold an ECVAA/ACVAA diploma and 3 were residents of the ECVAA at the time. We believe the definition is clear enough and the term ‘experts’ is classically used in the Delphi method to refer to the participants. We defer to the editor if he rather included further details in the results section about the population of participants or not.

a. participants may have provided limited descriptors, reflecting limited training experience. To some extent this is reflected in the Discussion, where the authors identify that some criteria, such as jaw tone, were not mentioned.

Please, see comments above.

b. participants reflect a single mother tongue, Spanish. It is well established that health assessment scale development should consider differences in the interpretation and meaning of terms between languages. It is concerning that a scale developed in Spanish has been translated in to English for publication, with no apparent assessment of whether scale performance has been affected as a result of translation.

The scale, the questionnaire, and the tool itself are exclusively translated to comply with the publishing guidelines of PLOS ONE or they are not translated at all. We are fully aware of language and cultural differences and we are not intending to validate this tool in English. That may occur at a further stage in the development of this project. Brondani, Luna and Padovani (2011) published the initial development of a scale for pain assessment in cats (originally in Portuguese) that finally unfolded into the ‘UNESP-Botucatu Multidimensional Composite Pain Scale for assessing postoperative pain in cats’, published and validated in several languages. This manuscript covers exclusively the initial development of a sedation assessment tool, nothing else.

c. Participant assessments using the developed scale were compared to a simple descriptive scale. This problematic because feedback from the same participants was used to develop the scale. Ideally, animals would have been video-recorded (with appropriate consent) and the the videos scored by a separate, independent group of observers.

As we described in the Material and Methods, the robust method of statistical analysis used in this study allowed us to use 66% of the data obtained to generate de statistical model (in this case the classification tree from the random forest), then this model was tested against the remaining 33% to confirm the validity of the model. In this article, we detailed the methods used to develop the sedation assessment tool, the next stage is to validate its inter- and intraobserver variability. This research is already in process and it will be published when complete. As you suggested, this phase is video based and it requires repeated assessment from multiple observers over a period of time.

d. the authors suggest that similarities in terminology and jargon used by participants during the Delphi process were beneficial. Given the lack of apparent minimal consistent training level, how could authors be certain that terminology and jargon were applied consistently by participants?

See above about training of participants. The terminology used during all the phases of the Delphi method is general medical/veterinary terminology. All the descriptor generated in the phase 1 and 2 were provided in the first submission.

2. Scale development process

a. the argument in favour of dichotomous outcomes as being more objective is misguided. Though forcing a participant to select a dichotomous outcome appears objective, the underlying assessments are in large part subjective. There is no indication that objective measures were employed in many of the final predictors described. This could explain why the scale performed less well when animals had levels of sedation that were not at the extremes (“no” or “profound” sedation).

It is not surprising that the tool performed slightly weaker between the mild and the moderate sedation. In many scales there is a considerable overlapping between the descriptors in these two levels and very often animals fit into descriptors from one or another category at the same time (forcing the assessor to take a subjective decision about what category to use). The composite scales developed until now have the same subjectivity in the score allocation. For instance, in Young et al. (1990), just in the only parameter with scores and descriptions – Spontaneous posture – the authors suggested using terms such as: standing/tired and standing/lying but able to rise/lying rising with difficulty. All assessment tools involved some degree of subjectivity and the combination of multiple dichotomic variables seems to provide the best results for the tool developed here.

b. from the methods (phase 1 and 2) it appears there was considerable author input in selecting and classifying the final descriptors - this may be a necessary step, but it would be helpful to have assurance that any author bias was limited - perhaps the authors would consider making their data available on-line?

The original response data from all the assessors and the summaries were included in the original submission. We have added now the involvement of the coordinators as a limitation of the study (line 345).

c. some descriptors are clearly highly subjective and uncontrolled e.g. responses to noise and name.

d. for some descriptors, participants had the option of selecting which would be tested. This suggests the possibility that the testing method was not standardised, with the risk that the outcome of a test would be influenced by what occurred previously. For example, if attempts were made to place a dog in lateral recumbency, this could change the level of sedation, affecting the test that followed (e.g. response to noise).

Please, see comments above about subjectivity.

e. There appears to be an underlying assumption that the method used by the authors to collect descriptors is somehow superior to methods used in previously published studies (e.g. Young et al.). Just because the described method has been named “Delphi” does not confer legitimacy over older studies conducted before the Delphi concept existed or was widely applied. It is highly likely that descriptors generated in previous studies were achieved through consensus discussion.

It is not the intention of the authors to start a discussion about the legitimacy of older or newer research, and we are sure that previously reported scales had a considerable amount of time dedicated to them. You appear to ignore the fact that 23 people giving their opinion about a topic and being asked for their opinion several times are likely to get a broader and more sound opinion (not necessarily better, although this is also likely) than three or four researches on their own. Additionally, the Delphi method is not a “new” method, this method (the Iman-Delphi Method) was first developed in the 40s in the early years of the Cold War to forecast the role of technology in warfare. Since then, it has been used in multiple areas and recently it is finding its way into areas where decision-making is necessary or fundamental.

f. I could not find Fig 2.

Figure 2 is at the end of the manuscript, before the links for the annexes. We have added to this letter for your convenience.

g. As identified by the authors, participants were not blinded to the status of the dogs. This would appear a critical limitation, especially when no further testing has been done to evaluate scale performance with observers blinded to treatment.

As we mentioned above, the robust statistical methods allowed us to performed this analysis. Additionally, as we already mentioned in the manuscript and in this letter, this research is just the initial development of the assessment tool and not the validation phase.

3. The language used throughout needs editing to raise the standard of scientific english. There is a tendency to use slang (“in fact”) and poorly defined terms (“awareness” - how can we know if an animal is aware? Similarly, “state of mind”).

This manuscript was written by a bilingual author and it was originally proof-read by a native English speaker. After your comments, the manuscript has been modified and it has been proof-read again to raise the standards of scientific English. It is difficult to address this criticism any further without particular suggestions.

4. As the authors describe, other sedation scales have been published for use in dogs, including one that was validated according to psychometric principles (validity and reliability testing). With the study goal of developing this novel scale, it would have been invaluable to perform some comparisons with pre-existing scale. In general terms, it is extremely difficult to interpret the performance of a novel scale when it is compared against a non-validated scale (as was the case here). Therefore, it is probably unsurprising that the results show the extremes of sedation (no sedation and profound sedation) to be in close agreement. Surely it is accurately classifying the full continuum (including crossing the threshold in to general anaesthesia) that is of greatest interest?

We appreciate your comment and agree that it would be interesting to compare several scales. As per the Material and Methods section, this research was performed before Wagner et al. (2017) was published. We will take into consideration your suggestion and, if possible, we might incorporate the comparison in future phases of the development of this sedation assessment tool.

We hope these comments answer all your queries and the changes in the manuscript address your concerns.

Once again, thank you for reviewing our work.

The authors.

Brondani, Luna and Padovani (2011) Refinement and initial validation of a multidimensional composite scale for use in assessing acute postoperative pain in cats. Am J Vet Res 72, 174–183.

Young LE, Brearley JC, Richards DLS, Bartram DH and Jones RS. Medetomidine as a premedicant in dogs and its reversal by atipamezole. J Small Anim Pract. 1990; 31: 554-559.

Wagner MC, Hecker KG and Pang DSJ. Sedation levels in dogs: a validation study. BMC Veterinary Research. 2017; 13: 110-118.

---

## [Decision Letter · Decision Letter 1]

14 Feb 2020

PONE-D-19-17102R1

The SIESTA (SEAAV Integrated Evaluation Sedation Tool for Anaesthesia) project: initial development of a multifactorial sedation assessment tool for dogs.

PLOS ONE

Dear Dr Martinez-Taboada

Thank you for submitting your manuscript to PLOS ONE. After careful consideration, we feel that it has merit but does not fully meet PLOS ONE’s publication criteria as it currently stands. Therefore, we invite you to submit a revised version of the manuscript that addresses the points raised during the review process.

Many thanks for your re-submission to PLOS One

Your manuscript was re-assigned to me as Academic editor.

Two reviewers were split on the manuscript, with one suggesting accept and one suggesting rejection

Therefore it went to a review by a third reviewer, and they have recommended that some changes be made.

I apologise for the delay in getting the comments back to you but hopefully you can understand why this was required.

I therefore invite you to further modify the manuscript, and resubmit it along with a response to reviewers comments

I wish you the best of luck with your revisions

Many thanks

Simon

We would appreciate receiving your revised manuscript by Mar 30 2020 11:59PM. To enhance the reproducibility of your results, we recommend that if applicable you deposit your laboratory protocols in protocols.io, where a protocol can be assigned its own identifier (DOI) such that it can be cited independently in the future. For instructions see: http://journals.plos.org/plosone/s/submission-guidelines#loc-laboratory-protocols

We look forward to receiving your revised manuscript.

Kind regards,

Simon Russell Clegg, PhD

Academic Editor

PLOS ONE

Reviewers' comments:

Reviewer's Responses to Questions

**Comments to the Author**

1. If the authors have adequately addressed your comments raised in a previous round of review and you feel that this manuscript is now acceptable for publication, you may indicate that here to bypass the “Comments to the Author” section, enter your conflict of interest statement in the “Confidential to Editor” section, and submit your "Accept" recommendation.

Reviewer #1: All comments have been addressed

Reviewer #2: (No Response)

Reviewer #3: (No Response)

2. Is the manuscript technically sound, and do the data support the conclusions?

Reviewer #1: Yes

Reviewer #2: No

Reviewer #3: Yes

3. Has the statistical analysis been performed appropriately and rigorously? 

Reviewer #1: Yes

Reviewer #2: I Don't Know

Reviewer #3: Yes

4. Have the authors made all data underlying the findings in their manuscript fully available?

Reviewer #1: Yes

Reviewer #2: Yes

Reviewer #3: Yes

5. Is the manuscript presented in an intelligible fashion and written in standard English?

Reviewer #1: Yes

Reviewer #2: Yes

Reviewer #3: No

6. Review Comments to the Author

Reviewer #1: Dear authors, thank you for revising the manuscript in a critical manner.

This paper is very interesting and it will be helpful for the clinical activity

Reviewer #2: Thank you to the authors for providing responses to my previous comments and suggestions. While I appreciate their efforts in justifying some of the decisions made during study design, I remain unconvinced that various assumptions underlying the design of this study are valid. Specifically, I do not feel that the justifications provided for the following are sufficient: wide variability in experience and training of participants, claims that performing the study in one language can generate a usable tool in another language, dependence on dichotomous outcomes, varying consistency in how assessments were applied by users.

Reviewer #3: This is an important and thorough investigation and it will be interesting to read further about the validation phase. However, I do have some doubts about the accuracy and clarity of English, which is a pity after the exhaustive research involved. To give some specific examples: frequent use of 'etc' when it's not clear about what the implied 'and others' actually are (e.g. a fundamental tool in the development of new sedative drugs, drug combinations, routes of administration, etc.); inconsistency in use of 'anaesthetists' and 'anaesthesiologists'; 'central nerve system'. I realise these are minor points, but I think they are vitally important because if we write descriptions of studies that might confuse readers assumed to be of equal knowledge and understanding, they might justifiably wonder whether the conduct of the study had been similarly confused. Maybe a good copy editor would be able to help, because this important work really needs to be published!

7. PLOS authors have the option to publish the peer review history of their article (what does this mean?). If published, this will include your full peer review and any attached files.

Reviewer #1: No

Reviewer #2: No

Reviewer #3: No

---

## [Author Response · Author response to Decision Letter 1]

19 Feb 2020

Regarding the specific comments from the Reviewers, please refer to the document named 'To the reviewers'

---

## [Decision Letter · Decision Letter 2]

10 Mar 2020

The SIESTA (SEAAV Integrated Evaluation Sedation Tool for Anaesthesia) project: initial development of a multifactorial sedation assessment tool for dogs.

PONE-D-19-17102R2

Dear Dr. Martinez-Taboada

We are pleased to inform you that your manuscript has been judged scientifically suitable for publication and will be formally accepted for publication once it complies with all outstanding technical requirements.

With kind regards,

Simon Russell Clegg, PhD

Academic Editor

PLOS ONE

Additional Editor Comments (optional):

Many thanks for resubmitting your manuscript to PLOS One

It was reviewed by the same reviewer as last time, and I am pleased to say that they have recommended publication

I have therefore recommended that your manuscript be accepted, and you should hear from the editorial office soon

I wish you all the best with your future research and it was a pleasure working with you

Many thanks

Simon

Reviewers' comments:

Reviewer's Responses to Questions

**Comments to the Author**

1. If the authors have adequately addressed your comments raised in a previous round of review and you feel that this manuscript is now acceptable for publication, you may indicate that here to bypass the “Comments to the Author” section, enter your conflict of interest statement in the “Confidential to Editor” section, and submit your "Accept" recommendation.

Reviewer #3: All comments have been addressed

2. Is the manuscript technically sound, and do the data support the conclusions?

Reviewer #3: Yes

3. Has the statistical analysis been performed appropriately and rigorously? 

Reviewer #3: Yes

4. Have the authors made all data underlying the findings in their manuscript fully available?

Reviewer #3: Yes

5. Is the manuscript presented in an intelligible fashion and written in standard English?

Reviewer #3: Yes

6. Review Comments to the Author

Reviewer #3: Thank you for making the amendments suggested in my last review. I think the manuscript reads much more clearly now.

---

## [Editor Report · Acceptance letter]

12 Mar 2020

PONE-D-19-17102R2 

The SIESTA (SEAAV Integrated Evaluation Sedation Tool for Anaesthesia) project: initial development of a multifactorial sedation assessment tool for dogs. 

Dear Dr. Martinez-Taboada:

I am pleased to inform you that your manuscript has been deemed suitable for publication in PLOS ONE. Congratulations! Your manuscript is now with our production department. 

With kind regards,

on behalf of

Dr. Simon Russell Clegg 

Academic Editor

PLOS ONE